# Advances in CAR T Cell Therapy for Non-Small Cell Lung Cancer

**Hong Yun Ma [1], Jeeban Das [2], Conor Prendergast [1], Dorine De Jong [3], Brian Braumuller [1], Jacienta Paily [1], Sophia Huang [1], Connie Liou [1], Anna Giarratana [1], Mahdie Hosseini [1], Randy Yeh [2] and Kathleen M. Capaccione [1,*]**

[1] Department of Radiology, Columbia University Irving Medica Center, 622 W 168th St., New York, NY 10032, USA; hym2103@cumc.columbia.edu (H.Y.M.); jpaily113@gmail.com (J.P.); mahdiehosseini1991@gmail.com (M.H.)

[2] Department of Radiology, Memorial Sloan Kettering Cancer Center, 1275 York Avenue, New York, NY 10065, USA

[3] RefleXion Medical Inc., Hayward, CA 94545, USA; ddejong@reflexion.com

\* Correspondence: kmc2113@cumc.columbia.edu

**Abstract:** Since its first approval by the FDA in 2017, tremendous progress has been made in chimeric antigen receptor (CAR) T cell therapy, the adoptive transfer of engineered, CAR-expressing T lymphocyte. CAR T cells are all composed of three main elements: an extracellular antigen-binding domain, an intracellular signaling domain responsible for T cell activation, and a hinge that joins these two domains. Continuous improvement has been made in CARs, now in their fifth generation, particularly in the intracellular signaling domain responsible for T cell activation. CAR T cell therapy has revolutionized the treatment of hematologic malignancies. Nonetheless, the use of CAR T cell therapy for solid tumors has not attained comparable levels of success. Here we review the challenges in achieving effective CAR T cell therapy in solid tumors, and emerging CAR T cells that have shown great promise for non-small cell lung cancer (NSCLC). A growing number of clinical trials have been conducted to study the effect of CAR T cell therapy on NSCLC, targeting different types of surface antigens. They include epidermal growth factor receptor (EGFR), mesothelin (MSLN), prostate stem cell antigen (PSCA), and mucin 1 (MUC1). Potential new targets such as erythropoietin-producing hepatocellular carcinoma A2 (EphA2), tissue factor (TF), and protein tyrosine kinase 7 (PTK7) are currently under investigation in clinical trials. The challenges in developing CAR T for NSCLC therapy and other approaches for enhancing CAR T efficacy are discussed. Finally, we provide our perspective on imaging CAR T cell action by reviewing the two main radionuclide-based CAR T cell imaging techniques, the direct labeling of CAR T cells or indirect labeling via a reporter gene.

**Keywords:** non-small cell lung cancer; NSCLC; CAR T cell therapy; imaging; solid tumors

## 1. Introduction

Lung cancer has been the leading cause of cancer deaths since the 1950s. The two main types of lung cancer are small cell and non-small cell carcinoma: non-small cell lung cancer (NSCLC) accounts for 85% of cancer related deaths. A total of 1.28 million new NSCLC cases were recorded during 2010 to 2017 in the US, at an annual incidence of more than 40 per 100,000 people and five-year survival of 26.4% [1]. Adenocarcinoma, squamous cell carcinoma, and large cell carcinoma are subtypes of NSCLC [2]. The different cancer subtypes are a result of mutations in different cells which ultimately undergo clonal expansion, resulting in a clinically apparent tumor. These differences in initial cell types result in a variety of clinical presentations and molecular signatures.

Cigarette smoking remains the largest risk factor for NSCLC. Other risk factors include sustained exposure to air pollution, occupational carcinogenic exposures, genetic susceptibility, and poor diet [3]. Nonsmoking-related lung cancer accounts for a significant

portion of lung cancer cases. Worldwide, 15–20% of men with lung cancer are non-smokers while over 50% of women with lung cancer are non-smokers [4]. There is an urgent need for effective treatments to treat and cure lung cancer.

Traditionally, NSCLC is treated with surgery, chemotherapy, and radiation therapy. These treatments can be combined to achieve the most effective outcomes. For cancer in stages I and II, surgical resection remains the most effective treatment [5], with adjuvant chemotherapy for stage II showing benefits. However, for stage I, there has been no proven benefit of adjuvant chemotherapy. More recently, immunotherapy and targeted therapy in combination with traditional treatment methods have significantly changed the treatment algorithm for patients with NSCLC.

## 2. Checkpoint Inhibitor Immunotherapy for NSCLC

Cancer cells are continuously being formed in our bodies but are efficiently eliminated by our immune systems in a systematic fashion. First, cancer antigens are released by dying cancer cells. Next, antigen-presenting cells such as dendritic cells recognize, process, and present the cancer antigens to T cells and activate them. The T cells then migrate to the remaining cancer cells, recognize them, and eliminate them through cell killing. However, some tumor cells can evade this immune surveillance through immune checkpoints. This phenomenon, known as the tumor escape mechanism, involves the recognition of tumor cells by surface antigen receptors normally expressed on T cells, including PD-1 and CTLA-4. When ligands from cancer cells (ex.: PD-L1) or ligands from antigen presenting cells (ex.: B7) bind to these T cell ligands, they cause an immunosuppressive response. For example, cancer cell ligand PD-L1 binds to T cell receptor PD-1 and causes the immunosuppression of T cells leading to deficient cellular immunity ("T cell exhaustion").

Thus, pharmacologic inhibition of these immune checkpoints aims to disrupt the interaction of their ligand/receptor pairs and thereby eliminate this immunosuppressive mechanism and restore cytotoxic T cell activity. This results in enhanced activation, proliferation, and T cell differentiation. Instead of directly killing the cancer cells, these pharmacologic agents enhance our immune system and its anti-tumor capabilities.

At least six checkpoint inhibitors have been approved for use in treating NSCLC. They are the anti-PD1 monoclonal antibodies nivolumab [5–7] and pembrolizumab [8–10], the anti-PD-L1 monoclonal antibodies atezolizumab [11,12] and avelumab [13,14], and the anti-CTLA-4 monoclonal antibodies ipilimumab [15,16] and tremelimumab [17].

The widespread use of checkpoint inhibitors, however, is also known to cause immune-related adverse effects [18]. The increased activation of autoreactive T-cells may be responsible for various immune-related adverse effects like autoimmune diseases. Commonly observed side effects include gastrointestinal toxicity, endocrine toxicity, and dermatologic toxicity. Others such as neurotoxicity, cardiotoxicity, and pulmonary toxicity are relatively rare but can be serious and even fatal.

## 3. Targeted Therapies for NSCLC

Targeted therapy offers a more precise approach to mitigating cancer cells than chemotherapy, immunotherapy, or resection. NSCLC has displayed mutations in receptors or protein kinases which initiate a cascade of cross-signaling pathways such as the RAS-RAF-MEK-ERK or MAPK/PI3K-AKT-mTOR or JAK-STAT pathways [2]. Multiple targeted agents have been developed to treat NSCLC. The targets include proteins along the signaling pathways involved in NSCLC or agents that fight the cancer cell antigens directly. Some of these targets include EGFR, ALK, MET, and HER2.

EGFR is a tyrosine kinase that, when activated, generates multiple signal transduction cascades. If mutated, EGFR can become dysregulated and lead to cellular proliferation, resistance to apoptosis, invasion, metastasis, and angiogenesis [6]. For tumors with an EGFR mutation, the first-line treatment option is an EGFR-tyrosine kinase inhibitor. There are currently three generations of EGFR-TKIs approved to treat NSCLC patients displaying EGFR mutations. First-generation EGFR inhibitors include gefitinib and erlotinib and although they are effective at disease control for 11–14 months, 60% of patients developed a second EGFR mutation for which the first-generation EGFR inhibitors were not effective [7]. Second-generation EGFR inhibitors include dacomitinib and afatinib and were developed to prevent mutation to a resistant clone. Subsequently, a third-generation EGFR inhibitor named osimertinib was developed, which works by sensitizing and antagonizing EGFR T790M-containing mutant NSCLC cells; it has demonstrated increased overall survival compared to first-generation inhibitors [19].

ALK is another target tyrosine kinase. Rearrangements in ALK result in dysregulation of the receptor and downstream cellular proliferation. Three generations of ALK inhibitors have been FDA-approved for use in patients with ALK-positive NSCLC. Crizotinib, a first-generation ALK inhibitor was shown to be superior to standard chemotherapy with either pemetrexed or dicetaxel, with a median progression-free survival of 7.7 months compared to 3.0 months in the chemotherapy group [8]. However, rapid resistance was found to develop within 1–2 years of treatment, thus leading to the creation of second-generation ALK inhibitors including ceritinib, alectinib, and brigatinib [9]. Of them, alectinib crosses the blood–brain barrier and can potentially treat brain metastases [10]. The third-generation ALK inhibitor lorlatinib is a reversible, potent ATP-competitive small molecule that is effective against all known resistant mutants [11].

Other targeted therapies for NSCLC include HER2 monoclonal antibody trastuzumab [20], and bispecific antibody amivantamab [21,22].

Molecular targeted radiotherapy is a novel type of therapy that conjugates a therapeutic radionuclide with a targeting molecule to direct radioactive particles directly to cancer cells. This focused radioactivity arrests tumor growth via DNA damage and subsequently activates the immune system. It can be combined with other immunotherapy agents for a more robust response [12]. Fibroblast activation protein (FAP) expressed on cancer-associated fibroblasts is a popular target expressed in cancer cells and not in normal cells. Preclinical data has shown this to be a promising strategy in lung cancer and other types of cancers [13].

## 4. CAR T Cell Therapy

CAR T cell therapy is the adoptive transfer of lymphocyte T cells expressing chimeric antigen receptors (CARs). CARs are fusion proteins which provide T cells with engineered specificity [14].

The first CAR T cell therapy was approved by the FDA in 2017 for patients with relapsed/refractory (R/R) hematologic malignancies. To date, six (second-generation) CAR T cell products are commercialized for certain types of leukemia, lymphoma, and myeloma; hundreds of clinical trials are in progress worldwide to translate this potent therapeutic option into one for patients with solid tumors [15–17].

### 4.1. CAR T Cell Design: Five Generations and Counting

CAR T cells are all composed of three main elements: an extracellular antigen-binding domain, an intracellular signaling domain responsible for T cell activation, and a hinge that joins these two domains. Antibody-derived single chain variable fragments (scFvs) are the most used antigen-binding domains, but the use of nanobodies (15 kDa) is likely to increase as they present several advantages like high solubility and stability, as well as excellent tissue penetration in vivo [23]. The length of the hinge region can be adapted to optimize the distance between CAR T cells and tumor cells, and the transmembrane domain anchors the CAR to the cellular membrane.

The five generations of CARs have been used in T cell therapy since its creation in the late 1980s, each with successive improvements in the intracellular signaling domain responsible for T cell activation. In the first-generation CARs, the intracellular T cell domain consisted of the CD3ζ chain [24,25]. The second- and third-generation CARs introduced one or two costimulatory molecules, CD28 and 4-1BB, respectively, to improve the persistence and efficacy of the CAR T cells [26,27]. They had enhanced T cell proliferation, cytotoxicity, cytokine secretion, and an increase in T cell life, and, for the first time, showed a delay in tumor growth in NSCLC cell lines [27]. As of now, the second-generation of CARs is the most widely used.

The fourth-generation CARs, also referred as TRUCKs or armored CARs, combine the expression of a second-generation CAR with a key inducible component such as a cytokine or chemokine such as interleukin-12 which increases the activation of the CARs [28]. By fine-tuning the pro-inflammatory cytokines which enable T cells, the fourth-generation CARs were more resilient within the tumor microenvironment (TME). The fifth generation is also based on a second-generation CAR with the addition of the intracellular domain of a cytokine receptor (e.g., IL-2Rβ chain fragment) [29].

In addition to the modifications of the intracellular domain of the CAR T cells, numerous new designs of the extracellular domain are also being investigated to improve CAR T cell therapy clinical outcomes. For instance, alongside the sequential or simultaneous administration of different CAR T cells to fight tumoral antigen escape, various multitargeted CARs were also designed such as dual (the infusion of two kinds of CAR T cells targeting different antigens), bicistronic (co-expressing two different CARs in a T-cell), tandem (two separate scFvs in tandem in a single CAR), and split and inhibitory CARs (iCARS) to increase the specificity of the transfused T cells and thereby mitigate toxicities [30,31]. In brief, dual CARs co-express two different CARs in one T cell whereas in tandem CARs two different scFvs are expressed in a single CAR molecule. Both designs behave such that activation can be triggered through the recognition of either antigen [30]. In split CAR T cells, the activation is split between a first-generation CAR and a chimeric co-stimulatory CAR (CCR). Both CARs need to bind their specific antigen for activation. Finally, iCARs possess a receptor for a self-antigen inhibiting the activation of a second CAR targeting a tumor-associated antigen [31].

### 4.2. The Different Steps of the Treatment and Mode of Action

CAR T cell therapy enhances T cell responses against specific antigen-expressing cells. CAR T cell therapy starts with apheresis to collect lymphocyte T cells from a patient or donor. These T cells are then activated, genetically modified (generally through a viral vector) to express an antigen receptor of interest on their membrane, and expanded ex vivo. While the CAR T cells are being manufactured, patients generally receive bridging therapy to slow disease progression, followed by lymphodepleting chemotherapy to prepare for CAR T cell infusion.

Once infused, CAR T cells recognize surface antigens independently from MHC restriction. When the CAR T cell binds to the antigen, it forms a so-called immune synapse, and resulting in tumor cell death. Activated CAR T cells predominantly kill tumor cells through the secretion of granzymes and perforin and or via the engagement of their membrane-bound tumor necrosis factor (TNF) family ligands with the tumors' receptors counterpart to induce apoptosis. Once activated, CAR T cells also multiply and release cytokines and other soluble mediators that stimulate the immune system and may be involved in further cell killing [32].

### 5. Use of CAR T Cell Therapy in Hematologic Malignancies

Hematologic malignancies have traditionally been treated with chemotherapy, radiation, immunotherapy, and bone marrow transplant. However, the mortality rates of hematologic malignancies have remained extremely high, with many patients requiring admission into the intensive care unit (ICU) within the first year of diagnosis [33]. Consequently, novel immunotherapies have been approved by the FDA to better treat these malignancies. CAR T cell therapy has revolutionized the treatment of relapsed and refractory B cell acute lymphocytic leukemia (B-ALL), non-Hodgkin lymphomas (NHL), and multiple myeloma (MM) [20].

CAR T cells target cell surface markers on malignant cells, specifically, CD19 was found on the surface of B cells and was the first biomarker targeted with CAR T cell therapy to treat B-ALL and NHL [20]. Nearly 90% of newly diagnosed NHLs in the United States are derived from B cells, with diffuse large B cell lymphoma (DLBCL) being one of the most common and aggressive types diagnosed [21]. As such, several FDA-approved CAR T cell therapies aimed at treating DLBCL.

Axicabtagene ciloleucel (Yescarta) was approved for the treatment of DLBCL in 2017 following the ZUMA-1 single-arm phase II trial [21]. This trial enrolled 111 patients, with 101 of the patients ultimately receiving the infusion. The overall response rate from the trial was 83%, with 58% of the subjects achieving complete remission. This therapy showed a durable response over the course of 2 years [21]. Other anti-CD19 CAR T cell therapies approved to treat DLBCL include tisagenlecleucel (Kymriah) and lisocabtagene maraluecel (Breyanzi) [21]. However, CD19 antigen loss is frequently observed and can lead to cancer relapse after therapy. Due to this phenomenon, bispecific CAR T cells that can recognize both CD19 and CD20 markers have been engineered to prevent antigen loss and achieve complete remission [20]. Another biomarker that can be targeted by CAR T cells is B cell maturation antigen (BCMA). BCMA is highly expressed in malignant plasma cells making it an ideal target in the treatment of MM. MM accounts for 10% of the hematologic malignancies in the United States but despite the advances made in the treatment of MM, most patients with MM relapse due to drug resistance [22]. Idecabtagene vicleucel (Abecma) and ciltacabtagene autoleucel (Carvykti) are two anti-BCMA CAR T cell therapies that have been approved for the treatment of MM [34]. With the success of CD19-, CD20-, and BCMA-targeted CAR T cells, many other targetable biomarkers are being researched to prevent antigen loss and better treat hematological malignancies.

Although CAR T cell therapy has been successful in the treatment of these hematological malignancies, they can have significant side effects and toxicities. One of the most common toxicities associated with CAR T cell therapy is cytokine release syndrome (CRS). CRS is characterized by the excessive release of cytokines following CAR T cell activation. Patients with CRS can present with fever, fatigue, myalgia, poor appetite, hypoxia, and hypotension. If not identified early, patients with CRS can progress quickly, leading to hemodynamic instability and multiple organ dysfunction [20]. Another toxicity associated with CAR-T cell therapy is immune effector cell-associated neurotoxicity syndrome (ICANS). ICANS can occur concurrently with CRS and can manifest as encephalopathy, seizures, and motor deficits [33]. Both toxicities are reversible if identified early and managed appropriately. One of the fundamental reasons for the side effects of this therapy is the immunogenic power of CARs. Therefore, the main challenge of CAR T therapy is the design of less immunogenic CARs.

## 6. Clinical Trials of CAR T Cells in Solid Tumors

Despite the significant advances achieved in the treatment of hematological malignancies, such as the use of CD19 CARs in leukemia, the use of CAR T cell therapy for solid tumors has not attained comparable levels of success [35]. Solid tumors present barriers to effective CAR T cell therapy that are not encountered in hematologic malignancies, including limited tumor trafficking, complex TME, the presence of immunosuppressive factors, and difficulty in precisely targeting cancer-specific antigens while preserving healthy tissues [36]. Solid tumors can be deep in the body, posing a challenge for T cells to reach them. Further, solid tumors have a distinct microenvironment that stems from the tumors' abnormal metabolism, characterized by increased interstitial fluid, hypoxia, and a reduced pH level; all of which severely impact CAR-T infiltration and survival, while also contributing to immune escape and adverse events [37]. Additionally, tumor cells recruit inhibitory immune cells like myeloid-derived suppressor cells (MDSCs) and regulatory T cells (Tregs) to disrupt the cytotoxic capabilities of effector T cells, and express ligands associated with immune checkpoints, like programmed death ligand (PD-L1/L2), that inhibit the immune response [38].

To date, a variety of innovative strategies have been employed to improve CAR T cells and overcome the unfavorable impact of the TME of solid tumors. These approaches involve modifying CAR T cells by knocking out the expression of PD-1 or coadministration of CAR T cells with checkpoint blockers that target PD-1, PD-L1, and CTLA-4 [39]. Even with these modifications, the overall response rate of CAR T cell therapy has been less encouraging in solid tumors compared to hematologic malignancies due to the heterogeneous expression of these inhibitory receptors and ligands within different tumors [40]. Combination therapy with chemotherapy [41], radiotherapy [42], or with other immunotherapy, like combining CAR T cells with TGF-β targeting agents, has yielded promising pre-clinical outcomes in sustaining their antitumor activity [43]. Furthermore, introducing CAR T cells directly into the tumor site offers an additional option to overcome barriers in their efficacy against solid tumors [44].

A growing number of clinical trials have been conducted to study the effect of CAR T cell therapy on solid tumors, targeting different types of surface antigens (Table 1). Ahmed et al. carried out a trial to evaluate the systemic administration of human epidermal growth factor receptor 2 (HER2)-specific CAR T cells in patients with glioblastoma. Among a cohort of 17 patients, the median overall survival was 11 months from the initial T cell infusion, 25 months from the time of diagnosis. The HER2-CAR T cell therapy was well tolerated and showed clinical benefits for patients with progressive glioblastoma [45]. A preclinical study focusing on glypican-3 (GPC3) CAR T cell therapy in patients with hepatocellular carcinoma demonstrated survival rates of 50%, 42%, and 10% at 6 months, 1 year, and 3 years, respectively [46]. Liu et al. conducted a phase I clinical trial to evaluate epidermal growth factor receptor (EGFR) CAR T cells in patients with metastatic pancreatic cancer. Among the 14 patients, eight attained stable disease and four experienced a partial response lasting for 2 to 4 months. Notably, patients with stable disease demonstrated reduced EGFR expression on tumor cells along with a shrinkage of metastatic lesions in the liver. The median overall survival for the entire group was 5 months [47].

In a phase 1 clinical trial conducted by Tchou et al., c-mesenchymal-epithelial transition factor (c-Met), a widely recognized molecule that is overexpressed in multiple types of cancer, was used as a target antigen. mRNA-transfected c-Met-CAR T cells were administrated intratumorally in patients with metastatic breast cancer. Immunohistochemical analysis after surgical removal revealed substantial tumor necrosis at the injection site, cellular debris, and a loss of c-Met immunoreactivity [48]. Another study by Chen et al. provided initial evidence regarding the effectiveness of anti-mesothelin CAR T cell therapy in ovarian cancer. In this study, disease stabilization was achieved in two out of three patients, with a progression-free survival of approximately 5 months [49]. CAR T cell therapy has shown limited efficacy in patients with metastatic prostate cancer [50]. However, a recent clinical trial using prostate-specific membrane antigen (PSMA) as the targeted protein has

demonstrated notable success in treating metastatic castration-resistant prostate cancer (CRPC), even at a low dose [51].

**Table 1.** Several targeted antigens utilized in CAR T cell therapy for solid tumors in clinical trials.

| Type of Cancer | Targeted Antigens |
|---|---|
| Glioblastoma | HER2, IL13Ra2, EGFRviii |
| Neuroblastoma | GD2, GPC2, CD171 |
| Lung cancer | MSLN, EGFR, FAP, CEA, PSMA, MUC1, ROR1 |
| Mesothelioma | MSLN, FAP |
| Breast cancer | c-Met, MSLN, HER2, GD2, CD44v6, MUC1, EpCAM |
| Gastric cancer | Claudin18.2, HER2, MSLN |
| Hepatocellular carcinoma | GPC-3, MSLN |
| Pancreatic cancer | MSLN, EGFR, CEA, HER2, PSCA, CLDN18.2, CD133 |
| Renal cell carcinoma | CAIX, AXL, ROR2, EGFR, MSLN |
| Colorectal cancer | TAG-72, CEA, NK2GD, GUCY2C, DCLK1 |
| Ovarian cancer | FRa, MSLN, MUC1, NKG2D, HER2, CD276, TAG72, MUC16, 5T4 |
| Prostate cancer | PSMA |

## 7. CAR T Targets Investigated in NSCLC

CAR T cells have shown great promise for NSCLC. Several target antigens with high specificity are being investigated alone and combined with strategies to overcome the barriers related to the specific TMEs of solid tumors [16]. Some of the most promising include epidermal growth factor receptor (EGFR), mesothelin (MSLN), prostate stem cell antigen (PSCA), and mucin 1 (MUC1). CAR T cells with multi-targeted combinations of these have also been tested, and may be more effective than individual targeting [52]. These targets and current data are detailed below, as well as novel targets currently in clinical trials (Table 2).

**Table 2.** New and emerging targets in clinical trials.

| Trial Target | Objective | ID |
|---|---|---|
| MUC1 | MUC1-targeting CAR T cells for advanced NSCLC; they are also engineered for PD-1 knockout, to further enhance CAR T cell longevity and cytotoxicity | NCT03525782 |
| | MUC1-targeting CAR T cells for advanced/metastatic solid tumors with a 3 + 3 dose design, in order to optimize a recommended phase II dosage | NCT05239143 |
| | Dose escalation study for CART-TnMUC1 cells, assessing for the safe dosing following lymphodepletion in breast, ovarian, pancreatic, and NSLSC | NCT04025216 |
| EGFR | EFGR-targeting CAR T cell trials in exclusively advanced NSCLC, modified by CXCR5 | NCT05060796, NCT04153799 |
| | EGFR-targeting CAR T cells with knocked-out TGFß receptors, to hopefully enable better CAR T cell penetrance of solid tumors | NCT04976218 |
| ROR1 | Assessing the safety and tolerance of dosing ROR1-targeting CAR T cells in patients with NSCLC or relapsed ROR1+ triple negative breast cancer; optimized dose will then be recommended as a phase II dosage | NCT05274451 |
| MSLN | Study assessing the tolerance and safety of MSLN-targeting CAR T cells in patients with solid tumors | NCT04489862 |

### 7.1. EGFR (Epidermal Growth Factor Receptor)

As its name suggests, EGFR is a tyrosine kinase receptor that transduces extracellular signaling for cell growth. Its amplification has been associated with multiple cancers, including lung adenocarcinoma, glioblastoma, and head and neck cancers [53]. Approximately 15% of NSCLCs express EGFR, making it a prime target for CAR T cell therapy [54].

A 2019 preclinical study examined the effectiveness of third-generation CAR T cells targeted against EGFRvIII, compared with control MOCK-CAR cells. Their use in mouse models significantly increased survival time and decreased the lung tumor burden [55]. Additionally, they resulted in the shrinkage of lung cancer metastases, suggesting a potential avenue for CAR T cell therapy in tandem with surgery or radiotherapy to eliminate residual cancer cells [55]. Another study by Li et al. reported similar results, as well as enhanced cytotoxicity when the CAR T cells were delivered intratumorally [56].

Early success with in vivo studies has resulted in EGFR-targeted CAR T cells' clinical trials for further assessment. A phase I study seeks to test CXCR5-modified CAR T cells against EGFR in advanced-stage NSCLC (NCT05060796). Therapy was well tolerated except for a grade 3–4 increase in serum lipase. Another study has similar aims and is attempting to optimize the dosing of EGFR CAR T cells while assessing for adverse events (NCT04153799). Several patients experienced grade 1–2 epithelial toxicities such as mucositis, desquamation, and gastrointestinal hemorrhage, while one patient developed acute pulmonary edema that responded to tocilizumab, an anti-IL-6 receptor antibody.

### 7.2. MUC1 (Mucin 1, Cell Surface-Associated)

The glycosylated membrane-bound mucin proteins can also be overexpressed in lung cancers; in normal bronchioles and pulmonary submucosa, MUC1 is not expressed at observable levels except on the luminal side [57]. Anaplasia from cancerous mutations causes MUC1 to be expressed, in lung adenocarcinoma in particular [57].

A 2019 study by Zhou et al. tested MUC28z CAR T cells against MUC1-positive tumors injected into mice. Flow cytometry and cytokine analysis revealed the CAR T cells recognized the antigens and became activated and lytic; they dramatically reduced the tumor burden in xenografted mice within 4 days of injection [58]. Follow-up studies also observed longevity in CAR T cell tumor toxicity, as the cells killed newly injected tumor cells up to 81 days post-treatment [58].

A pilot study in humans sought to combine MUC1 CAR T cells with PD-1 knockout T cells to treat advanced NSCLC. It demonstrated significant primary tumor reduction, but less encouraging results on metastases (NCT03525782). Additional trials are attempting to replicate and expand on this success with MUC1 CAR T cells targeting solid tumors and metastases in the lungs, liver, pancreas, and breast tissue (NCT02587689 and NCT05239143).

### 7.3. MSLN (Mesothelin)

Mesothelin is an immunogenic glycoprotein abundantly expressed in NSCLC and mesothelioma cells, with no appreciable expression in healthy tissue; it is correlated with a poor prognosis and resistance to chemotherapy [59]. This makes it a prime target for CAR T cells. Experiments with second-generation MSLN-targeting CAR T cells have shown their therapeutic potential to inhibit tumor growth, though they were not able to completely eliminate the cancer [59].

To date, clinical trials have not yielded promising results. A phase I trial investigating MSLN CAR T cells' efficacy was terminated given that many patients experienced disease progression; there were also six cases of severe adverse events, and one death from the highest dose of CAR T cells (NCT02414269). Other clinical trials for MSLN-targeted CAR T cell therapy remain ongoing; one is testing the safety of different CAR T cells in mesothelioma, along with an anti-PD1 agent (NCT04577326). The potential enhancement of CAR T cell migration is being investigated in phase I/II trials with CD40 ligand-expressing MSLN CAR T cells (NCT05693844).

### 7.4. PSCA (Prostate Stem Cell Antigen)

PSCA overexpression was originally detected in prostate cancer, but has also been found to be frequently amplified in gastric, gallbladder, pancreatic, and lung cancer [60]. It is a small glycophosphatidylinositol-anchored cell surface protein; its function is not unknown. Antibodies against the molecule have inhibited the growth of tumors in mouse models but have also been oncogenic in gastric cancer cell lines [61,62]. This indicates that PSCA function is cell type- and context-dependent, warranting further investigation.

Multiple lines of CAR T cells targeting PSCA have been tested, often with CAR T cells against MUC1, for potential synergy. Third-generation CAR T cells against PSCA have delayed NSCLC development in patient-derived xenograft models [60]. A study by Wei et al. demonstrated the cytotoxicity of PSCA-directed CAR T cells against NSCLC. The addition of MUC1-targeted CAR T cells also showed a synergistic effect, with the overall tumor mass significantly less than in those treated with either MUC1- or PSCA-directed CAR T cells individually [60].

Multiple trials are investigating PSCA CAR T cells' effect on solid tumors, although there is a paucity of trials specifically investing lung cancer. One ambitious trial seeks to test a battery of CAR T cell targets—PSCA, MUC1, TGFß, and GPC3—against advanced-stage lung cancers (NCT03198052). But most studies of PSCA-targeted CAR T cells' effects are on prostate cancer, particularly in metastatic and castration-resistant cancer (NCT05805371). One noteworthy trial was suspended due to a dose-limiting toxicity effect of the CAR T cell therapy. (NCT02744287)

## 8. New Targets on the Horizon

In the pursuit of the enhanced efficacy of CAR T cell therapy for NSCLC, there have been numerous research efforts that have shown the promising effects of potential new targets. Several targets have been identified and are currently under study in clinical trials. These targets are erythropoietin-producing hepatocellular carcinoma A2 (EphA2), tissue factor (TF), and protein tyrosine kinase 7 (PTK7) (Table 3).

EphA2 is a promising target for CAR T cell therapy for NSCLC due to the distribution of this antigen. By using immunohistochemical methods in surgically resected NSCLC tissue samples, it has been discovered that EphA2 is overexpressed in more than 90% of NSCLC samples. A correlation of EphA2 expression and clinical factors showed that increased EphA2 expression correlated with a history of smoking, poorer prognosis, and decreased survival [63]. Given these results, EphA2 is a valuable tumor-associated antigen target for CAR T therapy in NSCLC. A research group has developed a CAR targeting EphA2 to investigate its efficacy in both in vitro flow cytometry and real-time cell electronic sensing system assays, and in vivo in a mouse model of NSCLC. They found that their CAR T therapy had specific cytotoxicity towards the tumor in their in vitro model and decreased tumor signals in their mice in vivo [64]. Future studies confirming these results and transitioning from preclinical studies to clinical trials may provide a promising direction for this research.

TF, also known as coagulation factor III, initiates the extrinsic pathway of the coagulation cascade and is a surface molecule that has been found to be overexpressed in certain cancers, including NSCLC. In vitro analysis of NSCLC tumors found that an increased expression of TF occurred in tumors that invaded blood vessels, highlighting its potential role in the metastatic process [65]. Studies examining the levels of TF in patients with NSCLC found that an increased expression of TF correlated with worse overall survival, and in vitro work showed that the knockdown of TF using siRNA inhibited tumor growth and metastasis [66]. These studies laid the groundwork for identifying TF as a potential tumor-associated antigen target for CAR T therapy in NSCLC. A recent study developed a third-generation CAR directed against TF. Using a mix of in vitro and in vivo models, including subcutaneous xenograft and lung metastasis models, they found that treatment with their TF-directed CAR T therapy suppressed the growth and metastasis of cancer cells that were TF positive. They also found no obvious toxicity by performing pathological

inspections of organs postmortem in mice that received the CAR T cell therapy [67]. These results are promising and highlight the potential for using TF as a targeted treatment for CAR T therapy in NSCLC.

PTK7 is a tyrosine kinase in the non-canonical Wnt signaling pathway considered a potential tumor-associated antigen target for CAR T therapy in NSCLC [68], due to its high expression in cancer stem cells in multiple solid tumors, including NSCLC. Work targeting PTK7 using a targeted antibody–drug conjugate containing humanized anti-PTK7 monoclonal antibody in conjunction with cytotoxic agents showed that PTK7-targeted treatment induced tumor regression that outperformed traditional chemotherapy in a mouse model of NSCLC using patient-derived xenografts in SCID mice [69]. Given these successes, researchers have worked on developing a CAR T therapy targeted towards PTK7. Using in vitro cytotoxicity assays, they found that their PTK7-targeted CAR T treatment was effective through multiple rounds of tumor challenge, and using in vivo xenograft mouse models, they found that their therapy suppressed tumor growth and increased overall survival. They also analyzed the levels of PTK7 expression in human tissues and found that, consistent with previous reports, there was no PTK7 expression in the major adult human organs. However, they did see low-level expression in some normal human tissues, including the digestive tract and kidney. These results show that CAR T cell therapy targeting PTK7 is a promising novel treatment for NSCLC, which may be given at higher doses than those in current clinical trials. However, care will have to be taken to evaluate off-target side effects in tissues such as those in the digestive tract [70].

## 9. Challenges in Developing CAR T for NSCLC Therapy

The efficacy of CAR T cell therapy has been well documented in hematologic malignancies. However, there has been less success with solid tumors with a poor response in a large proportion of patients on a single therapy. This is often attributed to the poor infiltration of T cells into the solid tumor microenvironment and T cell exhaustion. Some microenvironments may deactivate CAR T cells altogether. For example, chemokine CXCL5 from solid tumors induces the secretion of enzymes and/or T regulatory cells, suppressing local T cell activation [71]. Alternatively, these microenvironments may recruit fibroblast and myeloid cells to overexpress extracellular matrix proteins, forming a mechanical barrier against CAR T cells [72].

Another significant challenge is the heterogeneity of the expression of tumor antigens in solid cancers. Whereas the ligands and antigens expressed in blood cancers are relatively homogenous, some antigens in solid tumors (such as EGFR and PSCA) can be expressed in healthy tissue, raising the potential for organ damage, which has occurred in some aforementioned clinical trials [72]. This so-called on-target off-tumor toxicity has led to the strategy of Boolean AND Gate logic CAR T cells. Such CAR T cells would be activated and initiate cytotoxicity only after binding to cells with a tumor antigen and no secondary, healthy antigen [73]. This strategy would necessitate the engineering of multiple T cell receptors and does not solve the problem of cancers' mutating target molecules or modifying expression. Clinical trials for these logic-gated CAR T cells are underway, hunting for cells that have enhanced CEA expression and have lost HLA-A*A02 expression to better enhance the killing of solid colon, pancreatic, and NSCLC tumors (NCT05736731).

T cell exhaustion results from repeated antigen stimulation leading to decreased cytokine secretion and the upregulation of inhibitory receptors. There is then an increased interaction of the upregulated PD-1 receptors on the CAR T cells with the PD-L1 receptors on tumor cells, leading to inhibition of T cell function [74]. If PD-1 or PD-L1 activity can be blocked either by systemic antibodies or by using PD-1 knockout T cells, then CAR T cell function can be rescued. There have been multiple studies in mouse models demonstrating an enhanced anti-tumor efficacy in PD-1-disrupted CAR T cells [74]. Human clinical trials are currently underway with preliminary data so far showing that injecting PD-1 knockout T cells is safe, with no grade 3–5 adverse effects [74]; efficacy data are still being collected.

High levels of CAR T cells within the tumor are needed to achieve a strong anti-tumor response. Studies have shown that using chemotherapy and radiation can increase T cell infiltration into the tumor by inducing a local inflammatory state. For example, in an in vitro mouse model of NSCLC, there was an increased efficacy of PD-L1 CAR T cells against low PD-1 tumor cells after local radiotherapy [75]. Another method to enhance the penetration of T cells into the tumor includes using oncolytic virotherapy to lyse cells, increasing release of immunostimulatory cytokines. In models of NSCLC, McKenna et al. used mesenchymal stromal cells to successfully deliver engineered adenovirus to tumor structures which augmented the CAR T cell response against tumors [76]. The authors proposed that this was due to the release of tumor antigens from lysed tumor cells leading to recruitment of additional T cells.

The FDA has recently released guidance documents, including the implementation of the Regenerative Medicine Advanced Therapy (RMAT) designation program, that are highly relevant to CAR-T cell therapy development [77]. However, there are major challenges in manufacturing beyond the scientific challenges discussed above. Due to the limited capacity to manufacture large quantities, CAR T therapy is very expensive. Unfortunately, the production process for the lentiviral vectors in producer cell lines is relatively inefficient due to the inherent toxicity of the vectors to the cells. Alternative methods for the introduction of the chimeric constructs are needed to increase production and reduce costs. The consistency of manufacturing of CAR-T cell products remains to be improved.

The CAR T cell manufacturing process typically takes 9 to 14 days, involving their activation, followed by viral transduction and expansion ex vivo. Recently, Saba Ghassemi et al. [78] reported a new method to manufacture CAR T cells that can generate them within 24 h, from T cells derived from peripheral blood without the need for T-cell activation or ex vivo expansion. The rapid manufacturing of CAR T cells may reduce production costs and broaden their applicability.

## 10. Other Approaches for Enhancing CAR T Cell Efficacy

In addition to finding novel targets for CAR T therapy, recent work has investigated how to make current CAR T therapies more clinically efficacious. To achieve this, methods have been employed to increase the bioavailability of the drug in the targeted regions where it is needed. Metastatic NSCLC often spreads to the brain, and once there, the prognosis for patients is poor. Once cancer has spread to the brain, the blood–brain barrier (BBB) is an obstacle that treatments must pass to reach their targets. One group has used CAR T cells that target B7-H3, shown to be effective against NSCLC in both in vitro and in vivo studies, with the co-expression of the CCL2 receptor CCR2b, which, as they demonstrated, improves the ability of this therapy to cross the BBB and treat metastatic NSCLC in the brain [79].

Another method that has been employed to increase the efficacy of CAR T cells against NSCLC is the remodeling of the TME via microwave ablation (MWA) that destroys liver tumors using heat generated via microwave energy. A recent study utilized microwave ablation (MWA) in conjunction with CAR-T cell therapy targeted against Axl, a receptor tyrosine kinase that is overexpressed in cancer. While the CAR-T Axl-targeted treatment showed moderate tumor regression, they found that the combination of microwave ablation and the therapy significantly enhanced tumor suppression in an in vivo NSCLC patient-derived xenograft mouse model [80].

Based on these early results, CAR T cell therapy for NSCLC is just beginning to expand clinically. While there are several clinical trials currently underway, many research groups are also investigating pre-clinical targets that show great promise. These studies are expected to demonstrate the efficacy of CAR T cell therapy for NSCLC, find targets with decreased off-target effects, and combine this therapy with others to optimize the efficacy of treatment regimens.

**Table 3.** Emerging targeted therapies being tested in clinical models.

| | Pre-Clinical Targets | Models Used | | PMID |
|---|---|---|---|---|
| Single Therapy | EphA2 | In vitro<br>Flow cytometry, real-time cell electronic sensing system assays | In vivo<br>Xenograft SCID beige mouse model of EphA2-positive NSCLC | 29132013 [64] |
| | TF | Flow cytometry, human tissue microarray, cytotoxicity assay | Human NSCLC s.c. xenograft mouse model, human NSCLC lung metastasis mouse model | 28055955 [67] |
| | PTK 7 | Flow cytometry, killing assays, recursive cytotoxicity assays | NSCLC using patient-derived xenografts in SCID mice | 34475869 [70] |
| Dual Therapy | B7-H3 + CCR2b | Flow cytometry, human NSCLC tissue microarrays | NSCLC xenograft models | 35443752 [79] |
| | Axl + MWA | Flow cytometry | NSCLC subcutaneous xenograft models | 36261437 [80] |

## 11. Imaging CAR T Cell Action

The inability to track CAR T cells in vivo presents a significant barrier to understanding the therapeutic mechanism of the CAR T cell treatment of solid malignancies such NSCLC [81]. Prior attempts to assess CAR T cell function using tumor tissue biopsy, quantification in blood, and conventional imaging, have failed to accurately reflect their survival, proliferation, and penetration of solid tumors [82]. Molecular imaging using radionuclides, however, can allow for the visualization and therapeutic monitoring of CAR T cells in solid tumors using positron emission tomography (PET) or single-photon emission computed tomography (SPECT) with computed tomography (CT) to provide both functional and anatomical assessment [83,84]. Whereas 18-F-fluorodeoxyglucose ($^{18}$F-FDG) PET/CT has been widely adopted for staging and to assess treatment response in NSCLC [85] (Figures 1 and 2), there is no established role to date for FDG in imaging CAR T cell action.

At present, there are currently two main preclinical radionuclide-based CAR T cell imaging techniques: the direct labeling of CAR T cells or indirect labeling via a reporter gene. Direct labeling refers to the passive labeling of T cells with small molecules or nanoparticles in vitro [86,87] and is a relatively straightforward process, requiring minimal cellular manipulation. Parente-Pereira et al. developed two CAR T cells, targeting MUC1 and ErbB, passively labeled with $^{111}$In-tropolonate to allow for the high-resolution real-time tracking of CAR T cells with SPECT/CT imaging, enabling the successful tracking of cell migration in vivo [88]. In addition, Weist et al. [89] used $^{89}$Zr-oxine to radiolabel CAR T cells which were administered to glioblastoma- and prostate cancer-bearing mice and subsequently visualized using PET/CT imaging and monitored dynamically via PET for up to 6 days [89]. Bhatnagar et al. used $^{64}$Cu-labeled gold nanoparticles to radiolabel CAR T cells which were then infused into mice and imaged using PET, demonstrating intense activity accumulation in the lungs at 10 min post-infusion, with activity in the liver and spleen gradually increasing subsequently, confirming the technique's potential utility in solid tumor imaging [90].

Despite the benefits of direct labeling, there are several obstacles to the clinical translation of in vivo CAR T cell imaging to solid malignancies, including limitations related to radiotracer half-life (as the duration of CAR T cell therapy treatment can last several weeks) and the potential toxic effects of the nanoparticles, which have not been fully explored [86]. An indirect labeling approach has several benefits, including a broader time window, the binding of viable cells only (ensuring that no signal is detected from non-living cells) [91] and no loss of signal loss post cell division, as progeny CAR T cells will also

express the reporter protein. Indirect labeling involves using symporter-based [68,92–94], receptor-based [95,96], antibody-based [97], and enzyme-based [98–100] techniques.

Volpe et al. evaluated the sodium–iodine symporter (hNIS) [94] using a PET probe $^{18}$F-BF4 to assess the trafficking of CAR T cells in triple-negative breast cancer models, and observed a marked difference in the retention of CAR T cells on PET/CT, which was confirmed ex vivo. Receptor-based reporter genes have also been utilized to image CAR T cell therapy, including the expression of somatostatin receptors (SSTR) [96] and PSMA [95]. Vedvyas et al. engineered native human T cells for tumor targeting and SSTR2 for reporting an estimate of the density of SSTR2-expressing CAR T cells infiltrating solid tumors. The authors revealed a biphasic CAR T cell expansion and contraction pattern almost matching tumor growth and destruction, providing a visual tool for quantifying T cell infiltration in solid tumors [96].

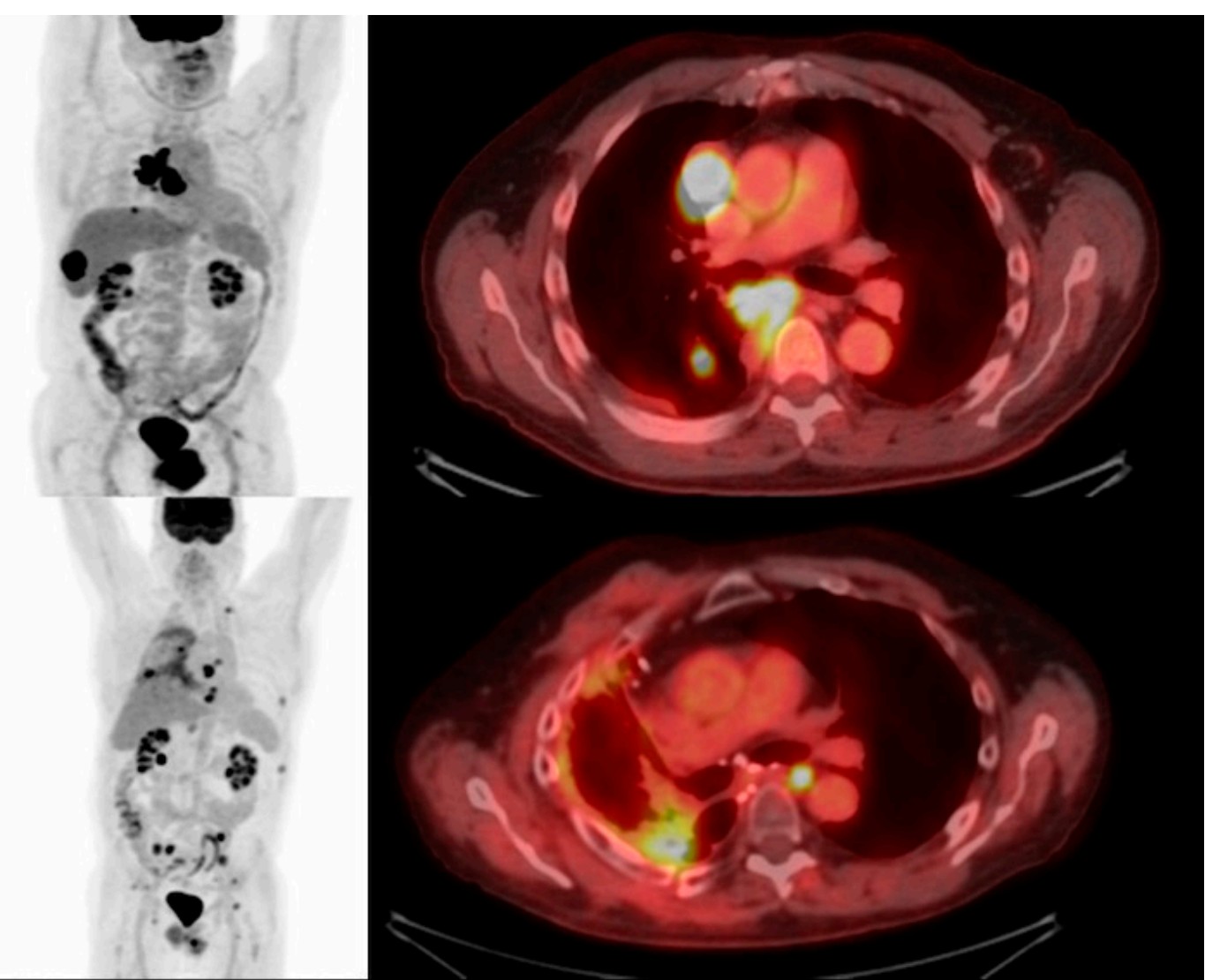

**Figure 1.** 68-year-old man with NSCLC. FDG PET/CT maximum intensity projection and fused axial image with FDG-avid thoracic nodal metastases and FDG-avid hepatic metastases. Subsequent PET/CT post systemic chemotherapy and radiation therapy and CAR T cell therapy with post treatment changes in the right lung demonstrates the post-treatment decreased mediastinal lymphadenopathy and primary tumor, however, new FDG-avid osseous metastases.

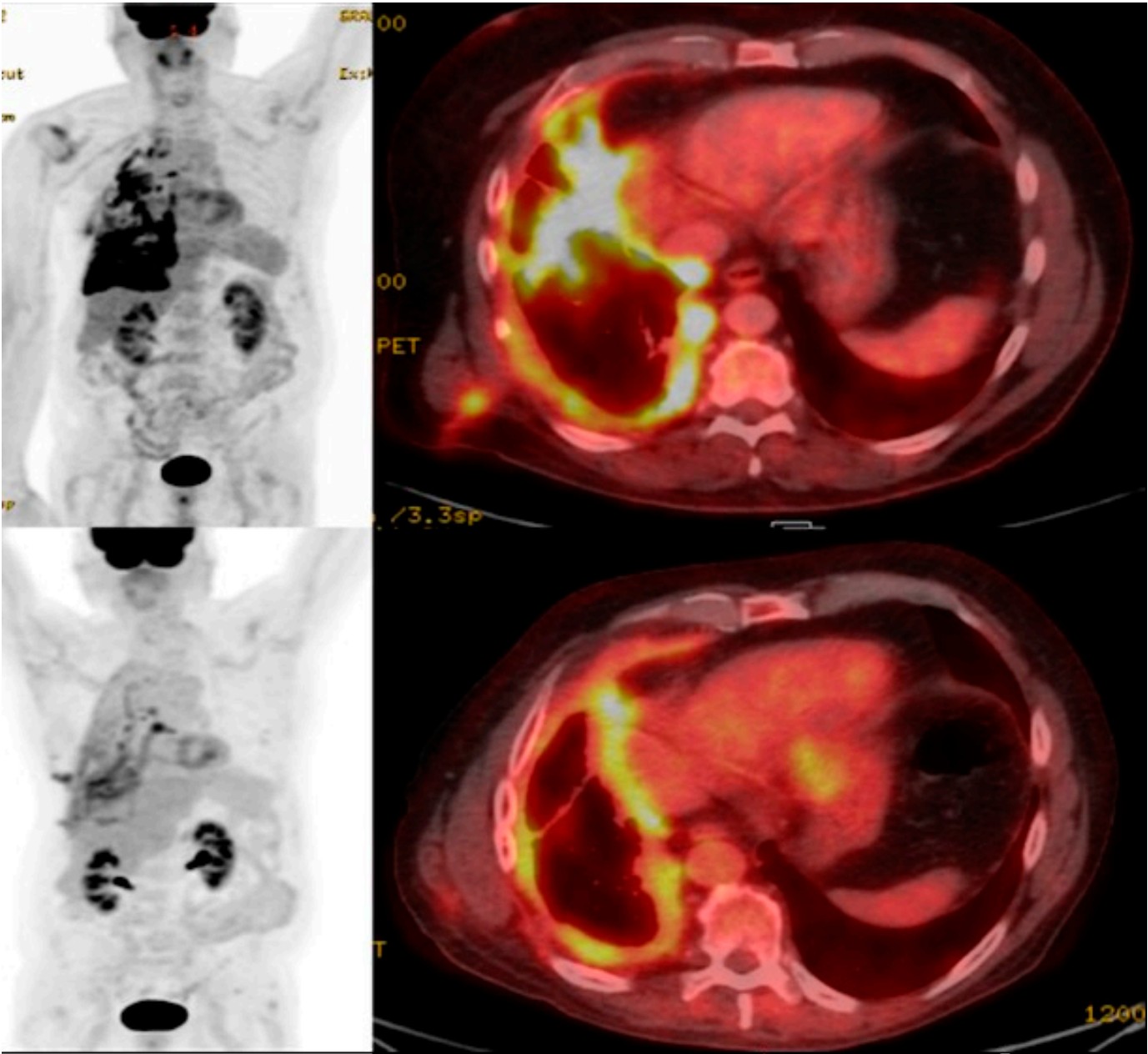

**Figure 2.** 68-year-old man with NSCLC. FDG PET/CT maximum intensity projection and fused axial image with FDG-avid right pleural metastases. Subsequent PET/CT post systemic treatment including CAR T cell therapy demonstrating decreased FDG-avid metastatic disease, consistent with treatment effects.

Subsequently, Minn et al. utilized $^{18}$F-DCFPyL PSMA to target and monitor CAR T cells in subcutaneous xenografts and osseous metastases in mouse models by inoculating them with CD19$^{+}$ Nalm6 cells and performing serial imaging to visualize CAR T cell infiltration in tumors [95]. The authors demonstrated that the number of CAR T cells in the peripheral blood and bone marrow did not correlate with tumor infiltration, indicating that bone marrow sampling may not provide effective monitoring of CAR T cells [95].

Krebs et al. [97] used DAbR1/AABD, a reporter probe pair comprised of a murine-derived single-chain antibody variable fragment bound irreversibly to lanthanoid-(S)-2-(4-acrylamidobenzyl)-DOTA (AABD) in vivo. AABD was then labeled with [86]Y and [177]Lu for PET/CT and SPECT/CT imaging, respectively, to track CD19-targeting CAR T cells infused either subcutaneously or intratumorally into subcutaneous U373 glioma xenografts. Both PET and SPECT showed that CAR-DAbR1 T cells aggregated in the high-CD19-expression tumors with favorable biodistribution and high image contrast, indicating DAbR1 reporter imaging with PET/SPECT can localize and track CAR T cells in solid tumors [97].

The only reporter gene currently used for CAR T imaging in human patients is an enzyme-based reporter gene, herpes simplex virus 1 thymidine kinase (HSV1-tk), which enables PET imaging with several radiolabeled probes, including [18]F labeled 9-(4-fluoro-3-hydroxymethylbutyl) guanine (FHBG) [98]. Yaghoubi et al. found that [18]F–FHBG can accumulate within glioma tumors [99], making transferred cells that express HSV1-tk detectable, and that it increased in sites of tumor recurrence, suggesting that CAR T cells migrated to the tumor sites, laying the foundation for further research into reporter gene imaging as a strategy to monitor the CAR T therapy of solid tumors, including NSCLC, in clinical practice.

## 12. Conclusions

Much progress has been made in treating NSCLC, the leading cause of cancer death. Traditional NSCLC treatment includes surgery, chemotherapy, and radiation therapy, and when combined these can achieve more effective outcomes. Recent developments have added immunotherapy and targeted therapies to the NSCLC treatment armamentarium. CAR T cell therapy, the adoptive transfer of lymphocyte T cells expressing chimeric antigen receptors (CARs), was first approved by the FDA for acute lymphoblastic leukemia. CAR-T cell therapy has revolutionized the treatment of hematologic malignancies; they now offer promise in NSCLC to further improve outcomes and generate a durable response. However, it remains a challenge to achieve effective CAR T cell therapy in solid tumors, due to the poor infiltration of T cells into the solid tumor microenvironment, T cell exhaustion, and the heterogeneity of tumor antigens' expression in solid cancers. Nonetheless, progress has been made towards addressing these issues, and there are emerging CAR-T cells that have shown great promise for NSCLC, such as epidermal growth factor receptor (EGFR), mesothelin (MSLN), prostate stem cell antigen (PSCA), and mucin 1 (MUC1). Potential new targets such as erythropoietin-producing hepatocellular carcinoma A2 (EphA2), tissue factor (TF), and protein tyrosine kinase 7 (PTK7) are currently under study in clinical trials. Imaging CAR T cell action will be increasingly important in the development of new CAR T cell therapies for NSCLC.

**Author Contributions:** Conceptualization, H.Y.M. and M.H.; writing—original draft preparation, all authors; writing—review and editing, H.Y.M. and M.H.; visualization, H.Y.M.; All authors have read and agreed to the published version of the manuscript.

**Funding:** This research received no external funding.

**Conflicts of Interest:** Author D.D.J. was employed by the company RefleXion Medical Inc. Author K.M.C. is an advisor for Cardinal Health. The remaining authors declare that the research was conducted in the absence of any commercial or financial relationships that could be construed as a potential conflict of interest.

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
