# Peer review of "Advances in CAR T Cell Therapy for Non-Small Cell Lung Cancer"

_cimb, doi:10.3390/cimb45110566_

Round 1

Reviewer 1 Report

Comments and Suggestions for Authors

Thank you for submitting this interesting and informative manuscript to Current Issues in Molecular Biology. I was pleased to receive it as a reviewer. While your manuscript provides valuable insights in an important clinical topic, there are some areas that could be refined to further augment the quality and impact of the work. Here are some respectful suggestions that could potentially improve the paper if you choose to implement them:

Abstract

- To make the abstract more accessible to general audiences, consider explaining acronyms, such as NSCLC, upon their first use.

Introduction

- To further enhance the contextual backdrop, you could include pertinent statistics regarding NSCLC incidence, mortality rates, and the 5-year survival prognosis. These statistics will serve to underline the gravity of the issue at hand and emphasize the pressing need for advancements in therapeutic strategies like CAR T cell therapy.

- Additionally, when discussing conventional treatment modalities for NSCLC, such as surgery, chemotherapy, and radiation, you could briefly mention the common side effects and inherent limitations associated with these interventions. This would not only elucidate the suboptimal outcomes associated with these traditional approaches but also underscore the exigency of exploring and improving alternatives, like CAR T cell therapy.

Checkpoint Inhibitor Immunotherapy for NSCLC

- The section on checkpoint inhibitors could benefit from the inclusion of a sentence addressing the frequency and severity of immune-related adverse events. This contextual information will lend a deeper understanding of the complexities and challenges associated with this emerging treatment modality and its potential risks, thereby fostering a well-rounded understanding of the therapeutic landscape in NSCLC.

CAR T Cell Therapy

- To augment comprehension of this section, you might consider including a table summarising the primary advantages associated with each CAR T cell generation. Such a table would succinctly capture the evolutionary progression, making it easier for readers to grasp the nuanced differences between these iterations.

- The explanation of the CAR T cell activation process is exceptionally clear. To further enhance reader comprehension, the addition of a visual aid, such as a graphic illustrating CAR T cell binding to tumour cells and the subsequent process of tumour cell elimination, would be a valuable addition. Visual representations can significantly assist readers in grasping complex processes.

Use of CAR T Cell Therapy in Hematologic Malignancies

- Since toxicity is a major concern with CAR T cell therapy, you could elaborate on the potential adverse events like CRS and ICANS. Expanding on these adverse events can help readers understand the potential challenges and risks associated with this therapy. In addition, discussing mitigation strategies would also be useful. Providing this information can help readers appreciate the efforts made to improve the safety and tolerability of CAR T cell therapy, enhancing the overall understanding of the treatment's clinical application.

- In the discussion of clinical trial results, a more detailed account of treatment responses would be advantageous. Providing specific data on the magnitude of treatment responses observed can ensure that readers are equipped with a more precise understanding of the clinical outcomes.

CAR T Targets Investigated in NSCLC

- In the context of selected antigen targets, such as EGFR and MUC1, it would be valuable to include information about any observed toxicities. This addition would ensure a balanced coverage of both the efficacy and safety aspects of CAR T cell therapy, allowing readers to gain a more comprehensive understanding of the therapeutic landscape.

- Discussing the immunosuppressive effects of TGF-beta could be a pertinent addition. This information would fit well within this context, as it would elucidate one of the key immunosuppressive mechanisms contributing to the complexities of treating solid tumours with CAR T cell therapy.

Conclusion:

- The conclusion effectively encapsulates the key points discussed throughout the manuscript and offers a forward-looking perspective. To enhance its impact, it is advisable to underscore the promise of CAR T cell therapy while also acknowledging the persisting challenges. Emphasizing the ongoing hurdles will help convey a more balanced perspective, leaving readers with a well-rounded understanding of the current state and future potential of CAR T cell therapy.

Overall, these suggestions aim to enhance the manuscript's quality and impact for clinicians and researchers. I believe that implementing some of the above suggestions would make your important work even stronger.

Author Response

Thank you very much for taking the time to review this manuscript. Please find the detailed responses in attached file and the corresponding revisions/corrections highlighted/in track changes in the re-submitted files.

Reviewer 2 Report

Comments and Suggestions for Authors

The topic is globally well presented and well structured. The writing of the text is very good. However, there are some detailed aspects that the authors could take into account to improve the manuscript.

In the abstract, the first time the acronym NSCLC appears (line 22) it should appear in parentheses to which words it corresponds.

In section 2 (Checkpoint Inhibitor Immunotherapy for NSCLC), it would be advisable to clarify that what is explained between lines 60-67 are known as tumor escape mechanisms.

In the paragraph before section 4.2 (lines 157-169) some CAR variants are discussed, the difference between dual CAR and tandem CAR should be better explained and the bicistronic model (which has not even been mentioned) should also be explained. Perhaps it would be necessary to make a graphic representation by means of a figure or scheme.

When between lines 221-231 some aspects related to the toxicity of CAR T therapy are discussed, it should be said that one of the fundamental reasons for the side effects of this therapy is the immunogenic power of CARs. Therefore, the main challenge of CAR T therapy is to design less immunogenic CARs.

In section 11 (Imaging CAR T Cell Action), different imaging techniques are cited to visualize morphologically and functionally the action of CAR T therapy. It would be advisable to also cite in vivo experimental studies (especially murine models) in which the luciferase activity of tumor cells is measured to monitor the action of CAR T therapy.

Author Response

Thank you very much for taking the time to review this manuscript. Please find the detailed responses in the attached file and the corresponding revisions/corrections highlighted/in track changes in the re-submitted files.

Reviewer 3 Report

Comments and Suggestions for Authors

This is a very well written review article focussing on the advances of CAR T cells in non small cell lung cancer. 

Following are some minor comments:

1. Authors should elaborate the regulatory guidelines (FDA or other) on acceptable practices during the development of CAR T cells. 

2. Authors should also highlight the process development including activation and number of days before the injection. 

Overall very nicely written article. 

Author Response

(The authors gave the same response as above.)
